# Effects of Microwave Treatment on Physicochemical Attributes, Structural Analysis, and Digestive Characteristics of Pea Starch–Tea Polyphenol Complexes

**DOI:** 10.3390/foods13172654

**Published:** 2024-08-23

**Authors:** Xin Zhang, Peiyou Qin, Dingtao Wu, Jingwei Huang, Jiayi Zhang, Yuanyuan Gong, Liang Zou, Yichen Hu

**Affiliations:** 1Key Laboratory of Coarse Cereal Processing of Ministry of Agriculture and Rural Affairs, School of Food and Biological Engineering, Chengdu University, Chengdu 610106, China; zxin1109@163.com (X.Z.); wudingtao@cdu.edu.cn (D.W.); huangjingwei@cdu.edu.cn (J.H.); zjiayi0626@163.com (J.Z.); gyy18784401268@163.com (Y.G.); zouliang@cdu.edu.cn (L.Z.); 2Institute of Agri-Food Processing and Nutrition, Beijing Academy of Agriculture and Forestry Sciences, Beijing 100097, China; qinpeiyou2020@163.com; 3Chengdu Agricultural College, Chengdu 611130, China

**Keywords:** pea starch, tea polyphenols, microwave treatment, physicochemical properties, digestive properties

## Abstract

Addressing the challenge of blood glucose fluctuations triggered by the ingestion of pea starch, we have adopted an eco-friendly strategy utilizing microwave irradiation to synthesize the novel pea starch–tea polyphenol complexes. These complexes exhibit high swelling capacity and low solubility, and their thermal profile with low gelatinization temperature and enthalpy indicates adaptability to various processing conditions. In vitro digestion studies showed that these complexes have a small amount of rapidly digestible starch and a large amount of resistant starch, leading to a slower digestion rate. These features are particularly advantageous for diabetics, mitigating glycemic excursions. Structurally, the pea starch–tea polyphenol complexes exhibited a B + V-shaped dense network with low crystallinity, high orderliness, and a prominent double helix content, enhancing its stability and functionality in food applications. In summary, these innovative complexes served as a robust platform for developing low glycemic index foods, catering to the nutritional needs of diabetics. It offers an environmentally sustainable approach to food processing, fostering human well-being and propelling innovation in the food industry.

## 1. Introduction

Starch, as the key carbohydrate storage form for plants, is commonly used by humans as a source of energy. Numerous studies have confirmed a strong connection between consuming starch and an increased likelihood of developing diabetes, heart disease, obesity, and cancer. Starch digestion can be categorized into three groups: rapidly digestible (RDS), slowly digestible (SDS), and resistant (RS). RDS is quickly broken down and absorbed efficiently in the small intestine, while SDS is metabolized at a slower rate. Resistant starch is not broken down in the upper digestive system but is instead used by the bacteria in the colon, which is crucial for keeping the colon healthy according to Englyst et al. [1]. The consumption of RDS can lead to significant fluctuations in postprandial blood glucose levels, posing potential health hazards. Therefore, low glycemic index starchy foods are important for diabetic patients and have been gaining considerable attention in recent years.

Pea (*Pisum sativum* L.), a globally significant food crop, with China being a key contributor to its global production, holds a pivotal position in the agricultural landscape. The primary constituent of pea seeds, known as pea starch (PS), comprises a substantial proportion of the dry seed mass, ranging from 24.7% to 51.2% [2,3]. Pea starch is noteworthy for its high amylose content, coupled with the high resistant starch content ranging from 7.9% to 22.2%. This unique composition endows PS with a high potential to be utilized as a dietary fiber for the creation of functional foods with a low glycemic index [2,4]. Pea starch has emerged as a promising candidate for use as a starch ingredient in the preparation of low-digestible starch. This has the potential to confer significant health benefits to humans.

Tea polyphenols, which are rich in catechins and derived from tea, have natural antioxidant properties and are safe for use in food, showing significant antioxidant effects [5]. Extensive research has shown that TPs significantly impact the physical and chemical properties as well as the in vitro digestibility of starch from various plant sources. Research has demonstrated that TPs can effectively reduce the aging process of rice starch through the use of sophisticated methods like differential scanning calorimetry (DSC) and X-ray diffraction (XRD) [6]. Furthermore, Zhao et al. verified the use of a dynamic in vitro rat stomach–duodenum (DIVRSD) model which showed that green tea polyphenols interact with lotus starch through non-covalent bonds in different ultrasound–microwave treatment settings, resulting in the creation of V-shaped inclusion complexes and non-inclusion complexes [7]. These interactions not only influence the physicochemical properties of lotus starch but also decrease its digestion efficiency. Conversely, the presence of (-)-epigallocatechin had no significant impact on the rate or degree of digestion of wheat starch, whether it was cooked or uncooked. However, the inclusion of (-)-epigallocatechin-3-gallate significantly reduced both the starch digestion rate and the degree of digestion [8]. The results indicated that adjusting the interaction between tea polyphenols and starch can help achieve specific functional characteristics in food systems.

Various techniques such as wet heat treatment, high hydrostatic pressure, microwave, ultrasound, dynamic high-pressure homogenization, and more have been employed to enhance the properties of starch [9,10]. Notably, MW irradiation technology, which falls under the category of non-ionizing radiation energy, holds significant potential in effectively altering the structural and functional attributes of food materials [11]. Microwaves are known for their efficiency and convenience compared to traditional heating methods. They allow for direct interaction with polar molecules in starch, resulting in changes to the granular structure through rapid heating and the “polar molecular vibration mechanism” [12]. During the digestion process, microwave-treated starch undergoes more significant molecular rearrangements compared to starch processed using conventional methods. For instance, the content of resistant starch in pure granular amylose starch undergoes a marked increase following extended microwave treatment [13]. Consequently, this method of processing results in a reduced digestion speed in the human gut, providing a distinct benefit in the field of food production [10].

Moreover, microwave power helps break down hydrogen bonds in starch molecules and influences the interactions between starch and polyphenols. Polyphenols are widely employed in food processing to mitigate the risks associated with various non-communicable chronic diseases. Instead of existing as standalone components, their unique molecular structure allows them to form complexes with starch. Although microwaves have been extensively employed for starch modification, no studies have been performed regarding microwave application for promoting V-type complex formation between pea starch and tea polyphenols. Additionally, the digestibility of the pea starch–tea polyphenol complexes and the mechanism by which microwave power promotes complex formation are not well understood. Herein, a proof-of-concept study to prepare the pea starch–tea polyphenol (PS-TP) complexes by green microwave technology, as well as investigate their structural properties and digestibility in vitro, was implemented. The present study would provide experimental support to develop a low GI product mediated by the green pea starch modification.

## 2. Materials and Methods

### 2.1. Materials

The pea starch (PS) was obtained from the company Yantai Shuangta Food Co., Ltd. (Yantai, China). Tea polyphenols (TPs) with a minimum purity of 98%, Folin–Ciocalteu reagent, and amyloglucosidase (AMG) with an activity level of 100,000 U/mg were acquired from Shanghai Yuanye Bio-Technology Co., Ltd. (Shanghai, China). Furthermore, Sigma Chemical Co. (St. Louis, MO, USA) supplied porcine pancreatic α-amylase (PPA) with an activity level of 50 U/mg. All other chemicals utilized in this study adhered to analytical grade standards.

### 2.2. Preparation of Samples

#### 2.2.1. Preparation of Samples Treated with Microwave

Around 10.0 g of pea starch and 1.0 g of tea polyphenols were dissolved in 150 mL of distilled water [14]. Afterward, the blend was moved to an MW reactor that had a cooling water circulation system (MKX-H1C1A, Qingdao Maikewei Microwave Innovation Technology Co., Ltd., Qingdao, China). The mixture was then treated with microwave irradiation ranging from 150 to 350 W for 8 min. In order to remove any simple polyphenols from the mixture, it was cooled to room temperature and rinsed three times with a solution of 50% ethanol and water. Subsequently, the mixture was freeze-dried for approximately 48 h [14]. The condenser temperature maintained during the freeze-drying process was approximately −80 °C. Finally, the samples were then milled and filtered through a 120-mesh sieve.

#### 2.2.2. Preparation of the PS and TP Physical Mixture

The physical mixture was prepared similarly to the microwave-treated samples, except that 10.0 g of PS and 1.0 g of TPs were manually blended for 8 min with a mortar and pestle.

### 2.3. Determination of TP Content in the Complex

The TP content was measured according to the Folin–Ciocalteu method and slightly modified [14]. Initially, 10 mg of the samples were suspended in 1 mL of deionized water and subsequently heated in a constant temperature water bath maintained at 90 °C for 10 min. After heating, the samples were subjected to centrifugation at 4000× *g* for 10 min followed by cooling. Subsequently, 0.1 mL of the supernatant was mixed with 0.5 mL of the diluted Folin–Ciocalteu reagent (one-tenth of the original concentration) using deionized water. The blend was kept at 30 °C for 5 min, then 0.5 mL of 20% (*w*/*v*) Na_2_CO_3_ solution was added and mixed thoroughly. Subsequently, the samples were shielded from light for 30 min, after which the absorbance was measured at 725 nm using a SYNERGYH1 Multifunctional Enzyme Marker (BioTek Instruments, Inc., New England, VT, USA). A standard curve was generated utilizing a range of gallic acid (GA) concentrations from 0 to 0.12 mg/mL, with an R^2^ value of 0.9985 and *n* = 7 replicates. The results were expressed as milligram equivalents of tea polyphenols per gram of dry weight of PS (mg GAE/g PS) [15].

### 2.4. Scanning Electron Microscopy (SEM)

The structures of the specimens were carefully examined with a scanning electron microscope (SEM, JLB-4700F, JEOL, Tokyo, Japan) to obtain a precise comprehension of their physical characteristics. The dried samples were securely affixed to an SEM stub using double-sided tape and spraying gold, ensuring that any floating powders were effectively removed. The entire process was conducted under a controlled low-vacuum environment, maintaining an acceleration voltage of 15 keV to optimize image quality. Micrographs were captured at magnifications of 2000×, 5000×, and 10,000× [16].

### 2.5. Laser Scattering Measurement

The laser diffraction particle size analyzer (LDSA, Malvern Mastersizer 2000, Malvern Instruments Ltd., England, UK) was utilized to accurately analyze the distribution of starch granule sizes, following the guidelines established by Jiang et al. [14]. The starch powder was dispersed within the analyzer cavity until an obscuration rate of 10–20% was achieved. Deionized water served as the dispersing medium, ensuring accurate particle size measurements. Polydispersity was analyzed using a refractive index of 1.33 for the water phase and 1.53 for the starch granules. The granule size reported represents the average of three replicated experimental measurements, ensuring the reproducibility and reliability of the results.

### 2.6. Wide-Angle XRD

The samples’ X-ray diffraction patterns were carefully examined with a Bruker AXS X-ray diffractometer (XRD, D8 ADVANCE, Karlsruhe, Germany), using Cu-Ka radiation at 40 kV and 40 mA. The measurement parameters were accurately adjusted to scan between 5 and 40 degrees with a step size of 0.02°, in accordance with the procedure outlined by Du et al. [16]. Subsequently, the degree of relative crystallinity was accurately determined using Origin Pro 2022 software, employing Equation (1) outlined below [17]:Relative crystallinity (%) = Ac/(Ac + Aa) × 100% (1)

The crystallinity area, denoted as Ac, refers to the area observed via X-ray diffraction, whereas Aa represents the amorphous area.

### 2.7. Fourier Transform Infrared Spectral (FTIR) Analysis

A Fourier transform infrared spectrometer (Spectrum Two, PerkinElmer, Waltham, MA, USA) was utilized to acquire spectrograms of the starches, adhering to the methodology outlined by Zhao et al. [7]. Precisely, 1 mg of the starch sample was mixed with 100 mg of KBr through thorough grinding. Afterward, the spectra were scanned across a span of 4000 to 400 cm^−1^, accomplished by accumulating 20 scanning times at a resolution of 4 cm^−1^.

### 2.8. Solubility and Swelling Power

The determination of solubility (S) and swelling power (SP) was conducted in triplicate using a modified protocol described by Guo et al. [18]. Initially, a starch sample equivalent to 0.6 g of dry weight was dispersed in 30 mL of distilled water within 50 mL centrifuge tubes. The tubes were then placed in a water bath and heated to temperatures of 55, 65, 75, 85, and 95 °C for 30 min. During this heating process, the tubes were mixed every 5 min using a turbine mixer (SCI-VS, Scilogex, C-6 Rocky Hill, CT (USA/Canada), USA) to ensure uniform heating and dispersion. Following the heating process, the samples were then cooled to ambient temperature and centrifuged at 4000× *g* for 15 min to separate the supernatant from the precipitates. The supernatant was carefully extracted using a pipette, while the precipitates were promptly weighed. Subsequently, the supernatants were subjected to a further drying process for 24 h at a temperature of 105 °C, after which they were weighed to determine the residual solids.

The solubility (S) was determined as a percentage, whereas swelling power (SP) was expressed in grams per gram of dry weight. These calculations were performed using Equations (2) and (3), respectively.
Solubility (%) = A/W × 100% (2)
Swelling power (g/g) = D/[W × (1 − S/100)](3)

Here, “A” represents the weight of the soluble fraction, “W” signifies the weight of the dried starch, and “D” denotes the weight of the swollen granules.

### 2.9. Thermal Properties

The gelatinization parameters of the complexes were meticulously evaluated using a differential scanning calorimeter (DSC 3500 Sirius, NETZSCH, Selb, Germany). Precisely weighted samples of 3.0 mg were directly dispensed into aluminum pans designed for DSC analysis. Afterward, 6 µL of distilled water was meticulously included in every sample. The pans were securely sealed and allowed to equilibrate at room temperature for 24 h, ensuring stability and reproducibility before analysis. The DSC scans began at a starting temperature of 20 °C. Subsequently, the temperature within the DSC unit cell was gradually raised to 95 °C, with increments of 10 °C/min, allowing for precise monitoring of the gelatinization process.

The onset, peak, and final temperatures (T*_O_*, T*_P_*, and T*_C_*) were precisely identified from the data collected, and the enthalpy of gelatinization (ΔH) was computed using the integrated peak area [19].

### 2.10. In Vitro Digestion

The digestibility of the pea starch–tea polyphenol complexes was thoroughly analyzed through the modification of previously established protocols [20]. Freshly prepared PPA and AMG solutions were employed for the digestion assay to ensure accurate results. A total of 100 mg of the specimens was mixed in a sodium acetate buffer solution (0.1 mol/L, pH 5.2, 15.0 mL) to create a uniform blend, then placed in a boiling water bath for 30 min. Subsequently, 10 mL of PPA solution (280 U/mL) and 1 mL of AMG solution (2500 U/mL) were added to initiate the digestion process. Afterward, the blend was transferred to a water bath set at 37 °C for 180 min, while being gently stirred at 120 revolutions per minute to ensure even digestion. Next, 0.5 mL samples were taken periodically and combined with 4.5 mL of 80% ethanol solution to stop enzyme activity and separate any starch that was not digested. The glucose content in the blend was precisely measured with the 3,5-dinitrosalicylic acid (DNS) technique [21]. Each sample was analyzed in triplicate to ensure reproducibility and statistical reliability.

RDS, SDS, and RS indicate the amounts of starch that hydrolyzed to glucose at different rates: RDS in less than 20 min, SDS between 20 and 120 min, and RS not breaking down after 120 min. The levels of RDS, SDS, and RS were measured using Equations (4)–(6):RDS (%) = (G_20_ − FG) × 0.9/TS × 100%(4)
SDS (%) = (G_120_ − G_20_) × 0.9/TS × 100%(5)
RS (%) = (TS − RDS − SDS)/TS × 100%(6)

The quantity of glucose released after 20 min is denoted as G_20_ (in mg), while G_120_ represents the mg of glucose released following 120 min. FG stands for the mg of free glucose present, and TS refers to the total starch content (in mg) of the complexes utilized for digestibility measurements. In this context, the TS value corresponds to 100 mg.

### 2.11. Statistical Analysis

Statistical analyses were conducted using the SPSS software package (version 22.0, SPSS Inc., Chicago, IL, USA). All figures and correlation analyses were generated using Origin Pro 2022 software (OriginLab Corporation, Northampton, MA, USA). The data from the experiment underwent a one-way ANOVA to evaluate its statistical significance. All test data were obtained in triplicate to ensure reproducibility and reliability of the results.

## 3. Results

### 3.1. Complexation Index of PS-TPs

The quantification of TP content in the complexes served as a metric for assessing the degree of complexation between PS and TPs, and the obtained results are shown in Table 1. Notably, the content of TPs in the physical mixture was low, amounting to merely 0.04 ± 0.01 mg GAE/g PS. This observation suggested that the physical mixture exhibited minimal binding affinity towards PS and TPs. Thus, the physical mixture was considered primarily composed of pea starch in this study. The content of TPs under varying microwave power conditions within the PS-TP complexes was found to be 8.40 ± 0.30, 9.30 ± 0.17, 16.76 ± 0.08, and 21.06 ± 0.51 mg GAE/g PS for PS-TPs-150 W, PS-TPs-200 W, PS-TPs-250 W, and PS-TPs-300 W, respectively. The findings suggested that the use of microwaves significantly improved the bonding between starch and polyphenols. Moreover, utilizing a microwave treatment and enhancing its power led to a notable rise in the concentration of attached TPs in the PS-TP compounds, far exceeding the quantities obtained through the physical mixture technique. Similarly, the previous studies to form the physical complexes between lotus seed starch and green tea polyphenols [7] or chlorogenic acid [22] were also reported. Nevertheless, when the microwave power rises to 350 W, a decrease in TP content was observed in the PS-TPs-350 W complexes, amounting to 18.53 ± 0.75 mg GAE/g PS. These results demonstrated that the optimal microwave power for achieving the highest degree of complexation between PS and TPs is 300 W. Increasing the microwave power beyond the optimal threshold was detrimental to the bonding process, as it impairs the effective formation of the complexes between PS and TPs.

### 3.2. SEM Analysis

The alterations in the microstructural morphology of PS-TP complexes resulting from the application of different powers of MW and the physical mixture (PM) were observed by SEM. As illustrated in Figure 1, a discernible distinction could be discerned between the morphology of the PM and PS-TP complexes. The starch of the PM displayed round and oval granules with a smooth surface, aligning with previous research on the shape of PS granules [23]. Following treatment by MW, the starch granules exhibited a visibly uneven appearance, showing altered shape and heightened surface porosity. These findings suggested that MW treatment has the capability to damage the surface integrity of PS. The damage degree of starch granules increased in line with the rise in microwave power. The phenomenon seen is likely due to the deep penetration of TPs into the starch granules when exposed to microwave radiation. This allows them to attach to the starch chains, breaking down the crystalline areas of starch and speeding up the breakdown of the original starch structure. This observation aligned with the results reported by Jiang et al. [14]. Furthermore, multiple aggregates adhered to the MW-treated PS granules, leading to an irregular distribution of particle sizes. These aggregates may be V-helical complexes formed by TPs attaching to the amylose that has been leached, using hydrogen bonding, hydrophobic interactions, or other types of forces. However, the creation of these complexes could impede the structured connections among the starch chains and damage the surface integrity of the starch gel, resulting in the development of fragments and irregular cavities [24,25]. Therefore, microwave treatment facilitates the formation of complexes between PS and TPs, while TPs accelerate the disruption of the initial structure of starch. The modifications could impact additional characteristics, leading to larger particle dimensions, paired helical structures, and enhanced swelling capacity, while reducing crystalline structure, solubility, and thermal stability.

### 3.3. Particle Size Distributions

Particle size distributions also reflect the morphological changes in the pea starch granules with tea polyphenols after being treated by microwave. The particle size distribution results by DLS analysis (Figure 2) and average particle size (Table 2) of PS-TP complexes were presented. As illustrated in Figure 2, compared to the physical mixture, the formation of PS-TP complexes resulted in an enhanced particle size. Taking the D [4,3] values of various groups for illustration in Table 2, along with the increase in microwave power, the D [4,3] values of the PS-TP complexes were basically risen. It would contribute to stronger power levels promoting aggregate formation and accelerating the interaction between PS and TPs. Interestingly, we observed a slight decrease in particle size of the PS-TP complexes at the 200 W microwave treatment, compared to the other counterparts. Actually, a similar phenomenon also has been found in previous studies [26]. Based on a previous study [7], because of the heat transfer within the granule during microwave treatment, the majority of starch granules would start swelling. Therefore, the particle size of PS-TPs at 150 W significantly expanded, compared to the physical mixture. However, when the power reached 200 W, a number of starch granules would break down [23,27], which resulted in the slight particle size shrinkage of PS-TPs [28]. Subsequently, the increased microwave power will cause more polymers derived from the broken starch granules to leach out, cross-link with each other, and form larger aggregates, because some amylose formed a V-type inclusion complex with TPs. As a consequence, when the microwave power exceeded 200 W, the average particle size of PS-TPs increased, along with the elevated power values. Furthermore, the formation of a large-particle-size complex between pea starch and tea polyphenols may result from the disruption of the orderly arrangement of starch molecules, thereby reducing the crystallinity of starch. The augmented particle size may influence the aggregation state of starch molecules, resulting in alterations to the starch surface area and specific surface area. The solubility, swelling power, and starch digestive properties could be affected by these modifications in an indirect manner.

### 3.4. X-ray Diffraction Spectra Analysis

The XRD patterns of the physical mixture and MW-treated PS-TP complexes are presented in Figure 3. The physical mixture of PS and TPs shows a characteristic C-type crystalline arrangement, confirmed by the strong diffraction peaks at 15.02°, 17.19°, and 22.98° in the XRD pattern [24]. Increasing the microwave power from 150 to 350 W led to a noteworthy change in the starch granules, causing them to lose their typical C-type crystalline structure and create B + V-type diffraction patterns. These diffraction patterns exhibited characteristic peaks at 2θ of 17.19° and 19.96°. A distinct peak around 17° indicated the presence of B-type crystalline structures [29]. Moreover, the diffraction peak at approximately 20° indicated the distinctive V-shaped structure, believed to be caused by the creation of complexes between pea starch and polyphenols. Amylose, a primary element in starch, has the capability to form a 6-fold left-handed single spiral structure. The spiral formations have the ability to connect and stack together, resulting in the creation of a crystalline structure known as the V-type [30]. The V-type crystalline structure exhibits hydrophilic properties on its exterior surface and a hydrophobic character on its interior spiral cavity. This distinctive structure enables the entry of small molecules into the cavity through hydrophobic interaction. Previous studies have shown that amylose is capable of creating inclusion complexes with various small guest molecules in the shape of V-type amylose complexes, which aligns with the current findings [31].

The level of relative crystallinity is a key factor in determining the extent of crystallization in a substance, with increased crystallinity typically associated with better thermal stability. The current research measured the relative crystallinity in the PM as 41.77%. The relative crystallinity of the PS-TP complexes showed a noticeable change following microwave treatment. An increase in power from 150 to 350 W resulted in a reduction in the crystallinity of PS-TP complexes from 13.23 to 10.02%, followed by an increase to 10.11%. The reduction in crystallinity was linked to the addition of TP molecules, potentially causing a disturbance in the organized arrangement and tight connection among the starch molecules, leading to a looser crystalline structure and nest-like structures. The SEM results also provided corroborating evidence for this decrease in relative crystallinity. However, at higher microwave power (350 W), the molecular motion became more intense, potentially triggering more complex structural rearrangements within the complex. This could include both the destruction of crystalline zones and possible recrystallization events, ultimately leading to an increase in relative crystallinity. The XRD data verified the formation of the PS-TP complexes, leading to the disruption of the starch’s well-organized structure. Additional examination was performed with FT-IR spectroscopy to study the structure with short-range order structure.

### 3.5. FT-IR Spectra Analysis

Fourier transform infrared (FT-IR) spectroscopy has been applied as an important tool to examine the subtle short-range molecular structural alterations in starch double helices [32,33]. Figure 4 displays the effects of the physical mixture and microwave treatment on the complex intermolecular interactions and changes in the crystalline and non-crystalline areas. All the samples exhibited a broad band between 3600 and 3000 cm^−1^, which was attributed to the stretching vibration of O-H in starch. The intensity of this band peaks at 3431 cm^−1^. The absorption peaks observed at approximately 2925 cm^−1^ and 2850 cm^−1^ referred to the asymmetric stretching vibration of CH_2_, while the band at 1418 cm^−1^ referred to the deformation vibration of CH_2_. Furthermore, the formation of PS-TP complexes was corroborated by the emergence of a novel peak near 2850 cm^−1^ in the microwave-treated PS-TP complexes. This observed phenomenon was primarily attributed to the interaction of specific functional groups present in tea polyphenols with the molecular structure of pea starch. Prior research has attributed this specific absorption peak to the asymmetric stretching and vibration of the CH_2_ bond [34]. Tea polyphenols contain groups such as methyl and methylene, which exhibit characteristic peaks related to C-H stretching and vibration in infrared spectroscopy. Upon interaction with pea starch, polyphenols form a complex where the C-H groups participate in intermolecular interactions with other functional groups along the molecular chain of pea starch. These interactions can lead to alterations in the vibrational modes within the corresponding spectral region, ultimately giving rise to the appearance of new peaks in the infrared spectrum. The 1640 cm^−1^ band was determined to be the C-O-O stretching vibration associated with a carbohydrate group. Additionally, the spectra reveal other characteristic absorption peaks at specific frequencies. The 1241 cm^−1^ peak corresponded to the -CH_2_OH-related mode, whereas the 1155 cm^−1^ peak was linked to the stretching of C-C and C-O bonds. The peaks at 1080 and 1020 cm^−1^ were linked to C-O stretching, and the peak at 929 cm^−1^ represented the skeletal mode vibration of the α-1,4 glycosidic linkage (C-O-C). The 855 cm^−1^ peak suggested deformation of C(1)-H and -CH_2_, whereas the 762 cm^−1^ peak was linked to the stretching of C-C bonds. The peaks at 709 cm^−1^, 575 cm^−1^, and 525 cm^−1^ correspond to the skeletal modes of C-C stretching and the pyranose ring, respectively [35].

The spectral bands detected at 1047 and 1022 cm^−1^ were determined to correspond to the amorphous and crystalline ordered areas. Furthermore, the band observed at 995 cm^−1^ was determined to be associated with the double helix structure, which supports the conclusions made by Luo et al. [36]. For a deeper understanding of the structural features of starch granules, the absorbance ratios at 1047/1022 cm^−1^ and 995/1022 cm^−1^ were computed to assess the level of short-range order (DO) and double helices (DD) [37,38]. The DO and DD values of different PS-TP complexes are shown in Figure 5a. A higher degree of short-range structural ordering is indicated by a higher value of DO, whereas a greater degree of molecular ordering is reflected by a higher value of DD [28]. The MW-treated complexes exhibited a superior degree of short-range structural ordering and a relatively high degree of double helices compared to the physical mixture. Furthermore, increasing the power to 300 W resulted in a significant improvement in the level of molecular order (DD). The observation suggested that increased power facilitated molecular organization and alignment. Several factors, such as increased power enhancing starch chain mobility, contribute to the favorable interactions between PS and TPs in this phenomenon. The increased movement of the starch molecules allowed for a more effective reassembly process, leading to the development of a more structured and aligned orientation [28]. Increasing the microwave power to 350 W can enhance the interaction between starch and tea polyphenols, possibly causing a partial disruption of the structured order and reducing the level of short-range order. Furthermore, exceedingly high microwave power can destabilize or disrupt the helical structure in starch molecules, ultimately resulting in a reduction in the degree of double helices.

### 3.6. Solubility and Swelling Power

The solubility reflects its level of dissolution, whereas the swelling power of starch serves as an indicator for evaluating its ability to absorb water [39]. The interactions between small molecules and starch chains in the amorphous and crystalline regions of the starch granule can be better understood by examining these two characteristics. The solubility and swelling power profiles of the physical mixture (PM) and microwave-treated PS-TP complexes are shown in Figure 5b,c. Both solubility and swelling power increased gradually as the temperature rose from 55 to 95 °C. As illustrated in Figure 5b, at temperatures of 55 and 65 °C, the solubility of the physical mixture was significantly lower compared to the MW-treated PS-TP complexes. Starch’s inability to dissolve is due to the strong hydrogen bonds that exist between starch molecules in both amorphous and crystalline regions. At a temperature reaching 75 °C, the pea starch within the water molecules begins to gelatinize. During this procedure, hydrogen bonds are formed between the water molecules and the hydroxyl groups present in the starch molecules. Consequently, the crystalline structure expands, causing the amylose to be leached out, resulting in a notable enhancement in the PM’s solubility [40,41]. However, compared to the PM, the solubility of the PS-TP complexes did not increase significantly. The existence of TPs and microwave treatment helped enhance the attachment of tea polyphenols to starch, leading to this outcome. This interaction gave rise to the formation of larger molecules or aggregates, as indicated by the observed particle size distribution. The formation of these aggregates would impede the connection between starch and water molecules, resulting in a reduction in the pea starch’s ability to dissolve. Meanwhile, the tea polyphenols themselves might also have competed with pea starch for water molecules, further influencing the solubility characteristics of the latter.

As demonstrated in Figure 5c, at 55 °C, the SP showed an upward trend as power increased, suggesting that MW could enhance the expansion of the starch granule. Meanwhile, the SP of all samples gradually increased with the increasing temperature. The swelling power of the PS-TP complexes also increased with increasing MW power. The SP value of the PS-TP complexes treated with MW was consistently higher than that of the physical mixture, suggesting the occurrence of gelatinization during the MW treatment. Studies have shown that the expansion of starch is mainly due to the existence of amylopectin, with amylose playing a role as a diluent, and a significant amount of amylose or binding molecules could help reduce the extent of swelling [42]. This suggested that amylopectin might undergo partial degradation into amylose because of MW treatment. The XRD results further demonstrated that TPs altered the crystalline structure of PS and reduced its crystallinity. The PS-TP complexes, which have lower crystallinity, probably have a looser structure and weaker intermolecular forces, leading to their better swelling properties.

### 3.7. Thermal Properties Analysis

DSC was used to determine the thermal characteristics of the physical mixtures and PS-TP complexes treated with microwaves, with the findings presented in Table 3. The melting temperature of the least stable crystallites, the predominant starch crystallites, and the most stable crystallites are denoted by T_O_, T_P_, and T_C_ values, respectively [43,44]. ΔH represents the energy necessary for melting the crystalline domains of starch [45]. The presence of TPs led to a notable decrease in temperature values (T_O_, T_P_, and T_C_) and ΔH values, with the extent of this decrease varying based on the power level of the MW treatment. The findings suggested that the amorphous and crystalline regions of starch suffered significant harm, leading to a decrease in the thermostability of starch in comparison to the physical mixture.

The previous studies have corroborated that complex polyphenols will remarkably change the thermodynamic properties of the gelatinization of corn starch [29] and rice starch [6]. The addition of polyphenols decreased the thermal stability of starch, resulting in an earlier destruction of the amylose lipid inclusion complexes. The possible reasons for the influence of polyphenols on starch gelatinization properties would be as follows [35]: polyphenols can interact with water molecules to change the pH value and ionic strength of the aqueous solution, thus changing the “surrounding environment” of the starch particles. The hydrophilic hydroxyl groups of polyphenols can interact with the side chain of amylopectin and bind to the amorphous region of starch granules to varying degrees, thus changing the coupling force between microcrystals and the amorphous matrix. In the present study, we could observe that although the microwave treatment will decrease the T_O_, T_P_, T_C_, and ΔH values of PS in comparison to the physical mixture, the different powers did not cause prominent differences among groups (Table 3). Differences among groups would be attributed to the microwave treatment, in which PS-TP complexes formed a strong interaction between TP and PS, which promotes the destruction of the starch microcrystalline structure and leads to changes in the T_O_, T_P_, T_C_, and ΔH of the PS. Although the different values of microwave power were employed, it may still be enough to ensure the relative uniformity of the internal temperature of the pea starch–tea polyphenol complex, so that the complex prepared at different powers shows small differences in thermal properties.

In other words, once the formation of a complex structure regardless of microwave power intensity occurs, the interaction between TP and PS may stabilize the structure. The gelatinization behavior of pea starch was restricted by tea polyphenols, showing relatively stable thermal characteristics.

### 3.8. In Vitro Digestion Characteristics

The widely utilized method for evaluating starch digestibility involves the use of pancreatic α-amylase and amyloglucosidase enzymes in the in vitro digestion process. This is because of its affordability and effectiveness [46]. This technique allows for the measurement of different starch fractions with varying digestion speeds, helping to gain a thorough comprehension of the digestion properties of PS-TP complexes under different MW powers. Figure 6 displays the impact of the PM and MW treatments on the in vitro digestion properties of PS-TP complexes, particularly the proportions of RDS, SDS, and RS. It is crucial to emphasize that the phenomenon being studied is not a unique characteristic or inherent quality of starch, but instead a kinetic indicator determined by the rates of enzymatic digestion [47]. The RDS, SDS, and RS levels in the physical mixture were 63.41 ± 0.33%, 5.35 ± 0.12%, and 31.24 ± 0.74%, respectively. When TPs were present, the microwave treatment greatly enhanced the levels of SDS and RS in the PS-TP complexes but reduced the amount of RDS significantly. It has been demonstrated that the introduction of TPs significantly enhances the resistance of carbohydrates to hydrolysis by enzymes. The impact was particularly significant when the microwave power reached 300 W, resulting in the PS-TP complexes showing the lowest RDS level of 41.67 ± 2.13%, with SDS and RS levels reaching 10.55 ± 0.36% and 47.78 ± 2.07%, respectively. The hydrolysis rate of starch is depicted in Figure 5b. The rates of digestion and hydrolysis exhibited by microwave-treated PS-TP complexes were notably diminished in comparison to that observed in the PM, indicating a substantial reduction in enzymatic activity towards these complexes. Current research aligns with prior studies that have shown how phenolic substances, such as tea polyphenols, flavonoids, and rutin, can inhibit starch digestion [48,49].

Polyphenols can slow down the digestion of starch by inhibiting amylase activity, which helps regulate blood sugar levels after eating foods high in starch. As digestion occurred, a notable quantity of TPs was discharged from the complexes, leading to a gradual decrease in catalytic effectiveness due to the occupation of the active sites of the digestive enzymes (α-amylase and/or amyloglucosidase) [50]. Disruption of enzyme activity can greatly affect the speed and amount of starch breakdown, resulting in changes to nutritional characteristics and bodily reactions. Moreover, the hydroxyl or phenolic groups present in TPs can establish additional hydrogen bonds with water molecules. The hydrogen bonds serve as a protective barrier for the amorphous regions of starch, preventing digestive enzymes from reaching them by limiting water access. The protective barrier reduces the hydrolysis of starch, resulting in a lower level of RDS and a slower overall digestion rate. Furthermore, polyphenols have the potential to affect the structural properties of starch molecules, thereby impacting their digestibility [6]. The application of microwave power enhances the migration of TPs within the starch matrix. This migration promotes the formation of intermolecular interactions between TPs and the starch chains. These interactions may result in structural modifications, including alterations in both the crystalline and amorphous areas of starch, which can impact its digestibility. Chai et al. previously conducted an investigation that revealed that the establishment of hydrogen bonding interactions between amylose molecules and TPs facilitates the clustering of amylose [48]. Subsequently, these aggregated structural units are dispersed within the amorphous regions of starch, giving rise to a denser network structure. Consequently, this network formation diminishes the ease of access of digestive enzymes to the substrate, resulting in starch exhibiting a distinct characteristic of delayed digestion.

### 3.9. Correlation Analysis

The unique structural characteristics and complex interactions among the constituent molecules of starch–polyphenol complexes are closely related to their functionality [28]. Figure 7 shows the analysis of the Pearson correlation coefficients between the multi-scale structural features of the PS-TP complexes and their digestibility characteristics to elucidate the cause of the variation in starch digestibility. This analysis covers various categories and their associated properties, including the complexation index (CI) and in vitro digestion parameters like RDS, SDS, and RS, as well as particle size distribution parameters such as D (10), D (50), D (90), D [4,3], and D [3,2]. It also considers indices of short-range structural order like DO and DD, relative crystallinity (RC), swelling power (SP), solubility (S), and the gelatinization peak temperature (T*p*). Correlation coefficients range from −1 to 1, with red circles representing positive correlations and blue circles representing negative correlations. The circles’ dimensions and colors indicate the strength of the positive or negative connections. They fall into the categories of low (|R| = 0.0–0.2), medium (|R| = 0.2–0.6), or high (|R| = 0.6–1.0), respectively. These classifications offer a quantitative assessment of the varying degrees of linear relationships present within the data. Significant changes were observed in the starch structure following microwave treatment, with starch digestibility showing a strong correlation with the complexation index (CI), short-range ordered structure, particle size, and swelling capacity. Notably, CI, particle size distribution, and resistant starch (RS) exhibited marked positive correlations. In particular, the elevated levels of CI (R = 0.97), D (10) (R = 0.49), D (50) (R = 0.90), D (90) (R = 0.91), D [4,3] (R = 0.91), and D [3,2] (R = 0.89) suggested that a rise in the extent of complexation and variation in particle size distribution were associated with an increased amount of RS. Moreover, the increase in swelling power (SP) (R = 0.93) and the degree of double helices (DD) (R = 0.79) suggested a strong connection between RS and swelling power. In contrast, RDS levels showed strong negative correlations with CI, particle size, DD, and SP, as well as a moderate negative correlation with DO, indicating that the creation of PS-TP complexes may result in decreased RDS content (R = −0.92).

## 4. Conclusions

The physicochemical properties of pea starch were examined after MW treatment with tea polyphenols. Structurally, the pea starch–tea polyphenol complexes showed a B + V-shaped dense network characterized by low crystallinity, high orderliness, and a significant double helix content. These complexes exhibited distinct physicochemical properties, encompassing high swelling power and low solubility, crucial characteristics for effective food applications. Its thermal behavior was characterized by low gelatinization temperature and enthalpy, indicating its suitability for various processing conditions. Furthermore, we found that the complexes contained only a small amount of rapidly digestible starch compared to a large amount of resistant starch, resulting in lower digestion rates. This property is particularly beneficial for individuals with diabetes, as it helps mitigate glycemic responses. In conclusion, this groundbreaking complex offers a strong base for creating low glycemic index (GI) products that cater to the dietary requirements of people with diabetes. Providing a sustainable and eco-conscious method for food processing that supports human well-being and fosters creativity in the food sector. Future studies will still need to explore the potential applications of these complexes in functional foods and their impact on glycemic control in individuals with diabetes.

## Figures and Tables

**Figure 1 foods-13-02654-f001:**
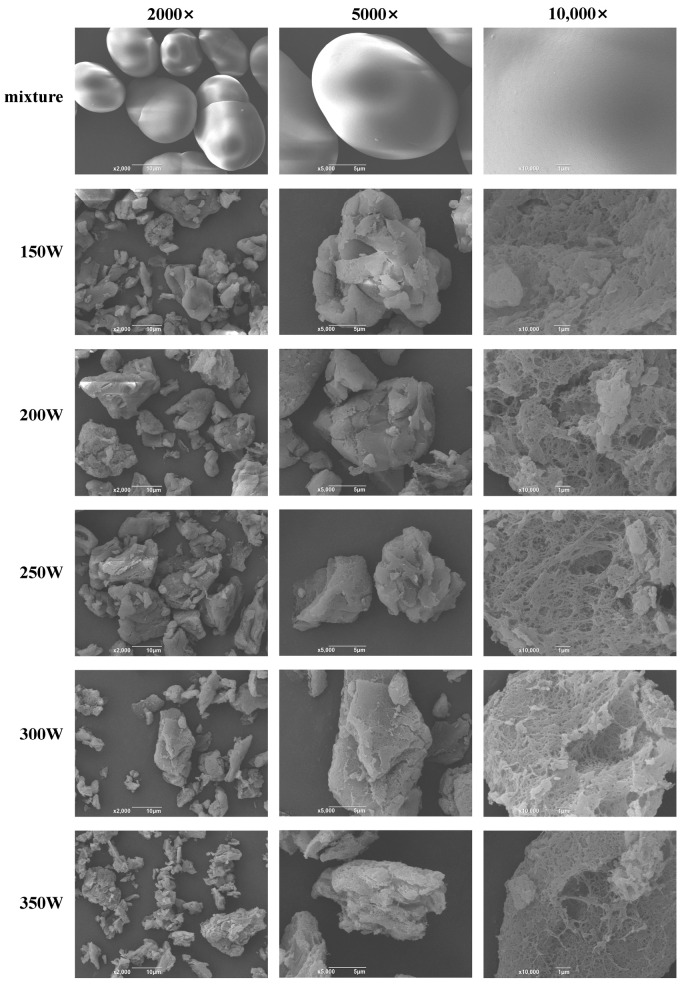
Scanning electron micrographs (2000×, 5000×, and 10,000×) of physical mixture and pea starch–tea polyphenol (PS-TP) complexes treated by different microwave powers (150 W, 200 W, 250 W, 300 W, and 350 W).

**Figure 2 foods-13-02654-f002:**
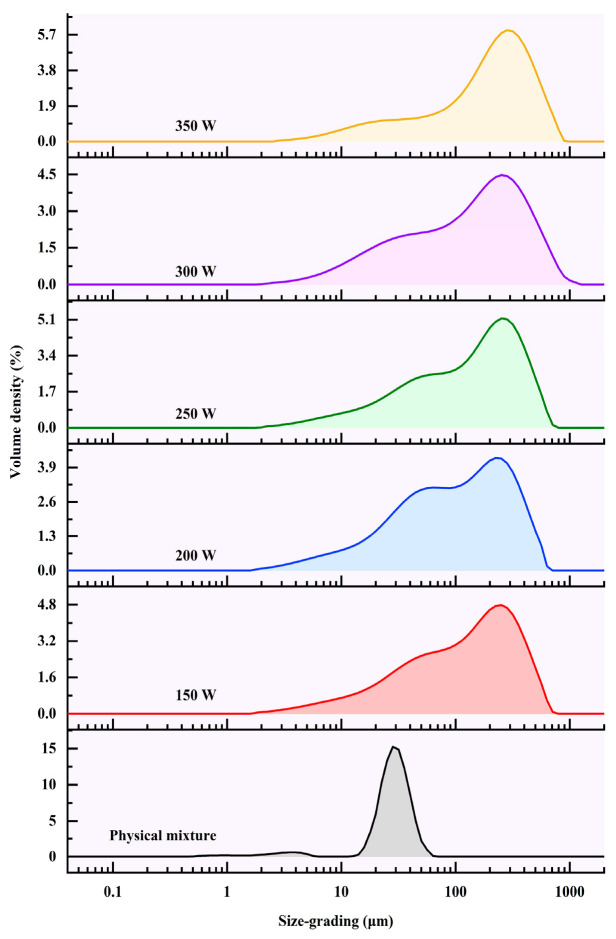
Particle size distributions of physical mixture and pea starch–tea polyphenol (PS-TP) complexes treated by different microwave powers (150 W, 200 W, 250 W, 300 W, and 350 W).

**Figure 3 foods-13-02654-f003:**
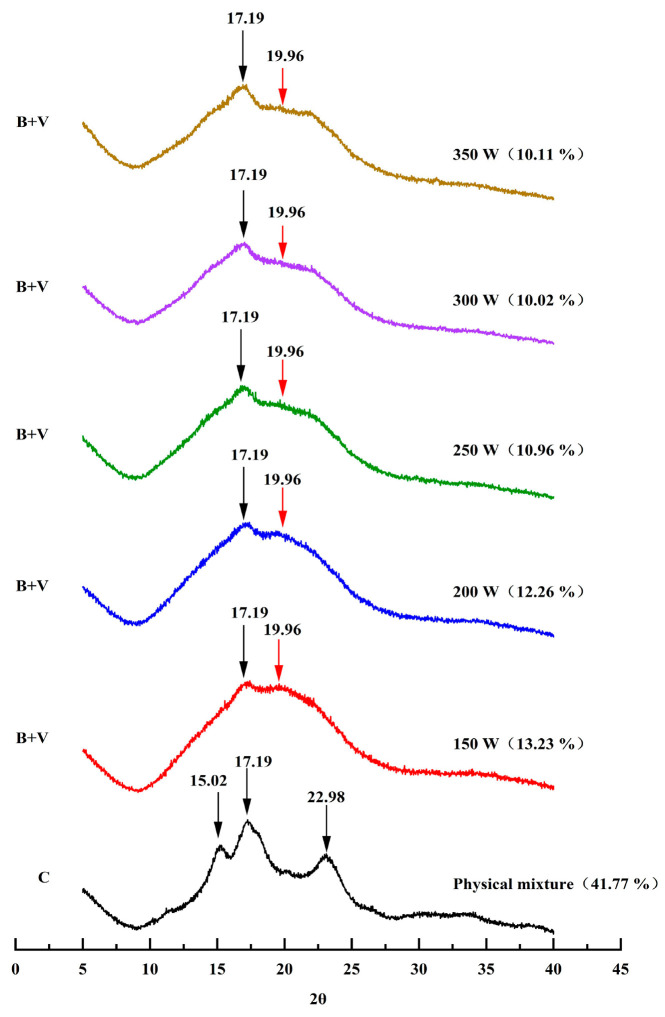
X-ray diffraction spectra of physical mixture and pea starch–tea polyphenol (PS-TP) complexes treated by different microwave powers (150 W, 200 W, 250 W, 300 W, and 350 W).

**Figure 4 foods-13-02654-f004:**
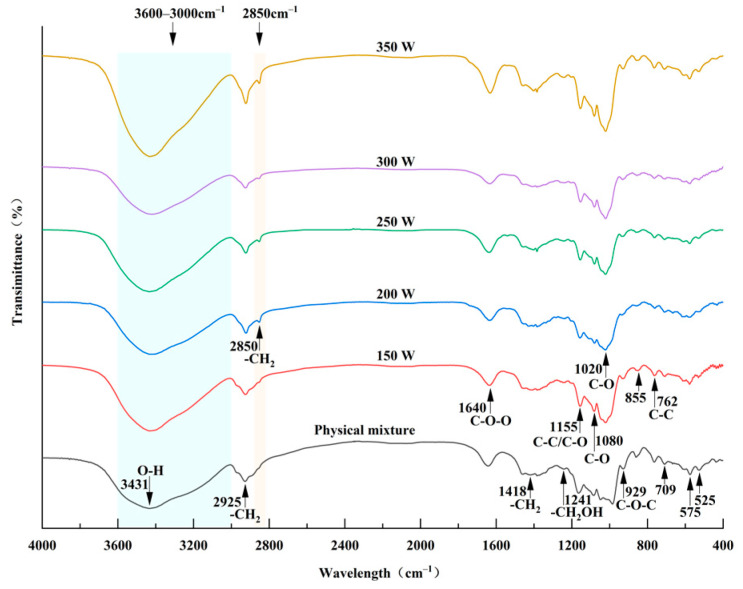
FT-IR spectra of physical mixture and pea starch–tea polyphenol (PS-TP) complexes treated by different microwave powers (150 W, 200 W, 250 W, 300 W, and 350 W).

**Figure 5 foods-13-02654-f005:**
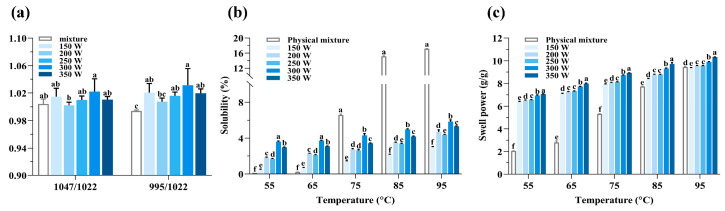
(**a**) The degree of short-range ordering (1047/1022) and the degree of the double helix (995/1022), (**b**) solubility, and (**c**) the swelling power of the physical mixture and pea starch–tea polyphenol complexes treated by different microwave powers (150 W, 200 W, 250 W, 300 W, and 350 W). Within the same column, values following a different superscript letter show significant differences (*p* < 0.05).

**Figure 6 foods-13-02654-f006:**
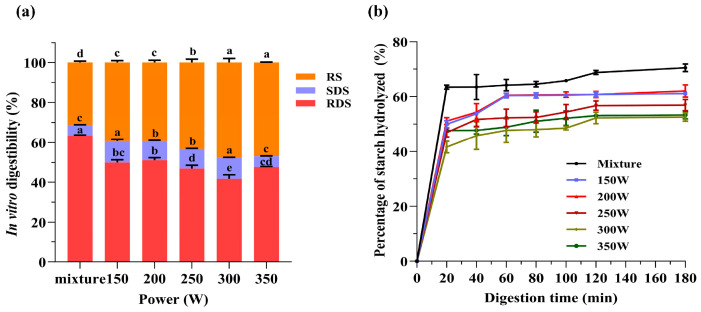
(**a**) In vitro digestibility and (**b**) hydrolysis rate of physical mixture and pea starch–tea polyphenol (PS-TP) complexes treated by different microwave powers (150 W, 200 W, 250 W, 300 W, and 350 W). RDS: rapidly digestible starch; SDS: slowly digestible starch; RS: resistant starch. Different letters in same row represent significant differences between different treatments (*p* < 0.05).

**Figure 7 foods-13-02654-f007:**
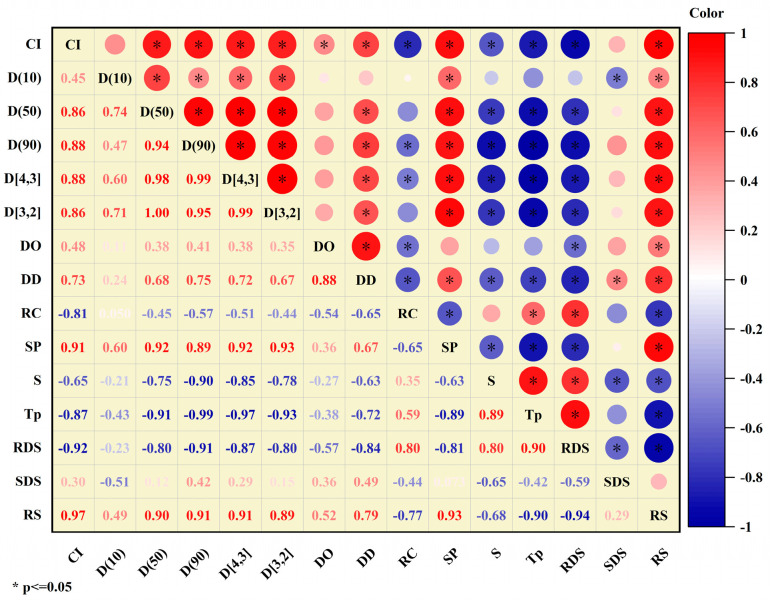
Analysis of Pearson correlation between in vitro digestion and structural properties of complexes (* *p* ≤ 0.05).

**Table 1 foods-13-02654-t001:** Complexation index of physical mixture and pea starch–tea polyphenol (PS-TP) complexes treated by different microwave powers (150 W, 200 W, 250 W, 300 W, and 350 W).

Power (W)	TP Content (mg GAE/g PS)
mixture	0.04 ± 0.01 ^f^
150 W	8.40 ± 0.30 ^e^
200 W	9.30 ± 0.17 ^d^
250 W	16.76 ± 0.08 ^c^
300 W	21.06 ± 0.51 ^a^
350 W	18.53 ± 0.75 ^b^

Experimental data are expressed as mean ± SD (*n* = 3). Within the same column, values following a different superscript letter show significant differences (*p* < 0.05).

**Table 2 foods-13-02654-t002:** Particle size distribution of physical mixture and MW-treated samples.

Power (W)	D [4,3]/μm	D [3,2]/μm	D (10)/μm	D (50)/μm	D (90)/μm
mixture	27.14 ± 0.20 ^f^	14.59 ± 0.13 ^e^	17.33 ± 0.25 ^d^	27.07 ± 0.08 ^e^	38.89 ± 0.77 ^f^
150 W	157.85 ± 0.72 ^d^	40.59 ± 0.26 ^c^	18.04 ± 0.18 ^c^	122.38 ± 0.85 ^c^	355.31 ± 2.27 ^d^
200 W	138.14 ± 1.71 ^e^	36.15 ± 0.33 ^d^	16.20 ± 0.17 ^e^	96.49 ± 2.01 ^d^	323.61 ± 3.24 ^e^
250 W	169.22 ± 0.33 ^c^	44.32 ± 0.04 ^b^	19.19 ± 0.01 ^b^	137.70 ± 0.67 ^b^	372.94 ± 0.36 ^c^
300 W	176.68 ± 0.34 ^b^	44.35 ± 1.30 ^b^	17.29 ± 0.33 ^d^	137.44 ± 5.50 ^b^	406.79 ± 0.61 ^b^
350 W	222.99 ± 4.41 ^a^	60.36 ± 0.71 ^a^	22.98 ± 0.34 ^a^	197.34 ± 2.14 ^a^	455.10 ± 5.03 ^a^

D [4,3]: mean volume diameter (μm); D [3,2]: weighted average of the surface area (μm); D (10), D (50), and D (90): the particles smaller than this diameter accounted for 10%, 50%, and 90% of the total number of particles, respectively (μm). Experimental data were expressed as mean ± SD (*n* = 3). Within the same column, values following a different superscript letter show significant differences (*p* < 0.05).

**Table 3 foods-13-02654-t003:** Thermal properties of PS-TP complexes formed by different microwave powers, compared to physical mixture.

Power (W)	T*_O_* (°C)	T*_P_* (°C)	T*_C_* (°C)	ΔH (J/g)
mixture	55.53 ± 0.93 ^a^	65.70 ± 0.82 ^a^	73.93 ± 4.81 ^a^	11.84 ± 0.86 ^a^
150 W	43.45 ± 0.35 ^c^	56.80 ± 1.13 ^b^	71.75 ± 0.49 ^ab^	3.04 ± 0.46 ^c^
200 W	41.90 ± 1.13 ^d^	56.95 ± 0.35 ^b^	68.27 ± 0.35 ^bc^	2.30 ± 0.25 ^d^
250 W	45.15 ± 1.06 ^b^	56.35 ± 1.06 ^b^	66.10 ± 0.99 ^c^	3.51 ± 0.24 ^c^
300 W	44.67 ± 0.60 ^bc^	54.83 ± 0.45 ^c^	64.57 ± 1.27 ^c^	3.23 ± 0.23 ^c^
350 W	43.70 ± 1.39 ^bc^	53.83 ± 0.55 ^c^	64.45 ± 2.98 ^c^	4.50 ± 0.28 ^b^

Experimental data are expressed as mean ± SD (*n* = 3). Within the same column, values following a different superscript letter show significant differences (*p* < 0.05).

## Data Availability

The original contributions presented in this study are included in the article, further inquiries can be directed to the corresponding author.

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
