# Peer review of "Effects of Microwave Treatment on Physicochemical Attributes, Structural Analysis, and Digestive Characteristics of Pea Starch–Tea Polyphenol Complexes"

_foods, 2024, doi:10.3390/foods13172654_

Round 1
Reviewer 1 Report
Comments and Suggestions for Authors
line 31 "reserved" - change word
line 46 "remarkable" - change word
The aim of the work was not clearly articulated, there is indeed a mention of what was not done/found in the available literature, but this is not the aim
very well planned scientific experiment. Chart 6 - very innovative graphic form presenting research results
Reviewer 2 Report
Comments and Suggestions for Authors
General comments
The manuscript entitled "Effects of Microwave Treatment on Physicochemical 2 Attributes, Structural Analysis, and Digestive Characteristics of 3 Pea Starch-Tea Polyphenols Complexes" shows very interesting results on the effect of using microwave radiation on mixtures of pea starch and polyphenolic compounds.
Specific comments
Authors must cite literature references reporting the effect of using microwaves on polyphenol molecules. Wouldn't just microwave radiation, at the frequency used in the study, cause degradation of polyphenols?
Line 58
Define the meaning of DIVR before the acronym.
Section 3.1. 3.1 Complexation Index of PS-TPs
In this section, authors must cite literature references that indicate the use of microwave radiation in the formation of complexes in starch molecules.
Lines 290 – 298
Although the argument is correct, the writing is somewhat confusing and/or monotonous. The authors should improve the presentation of the text. For example: there is no point in reporting all the same values ​​in the text that were already reported in table 2.
Lines 301 - 304
The authors must try to find a hypothesis for the different behavior at the frequency of 200 W compared to the trend of increasing particle size for the other frequencies.
Figure 2.
The authors must to split the x-rays spectra of the particle size distribution in two figures.
Table 3
Although the change caused by the use of microwave radiation in the thermal properties of the systems was clearly observable, the effect of the frequency change was relatively small on these same properties. The authors must present a hypothesis and/or other citations from the scientific literature for this behavior.
Table 4 and Figure 5a are repeating themselves. Authors must choose one of the ways to present the results only.
